# A Picture Is Worth a Graph: A Blueprint Debate Paradigm for Multimodal Reasoning

## ABSTRACT

This paper presents a pilot study aimed at introducing multi-agent debate into multimodal reasoning. The study addresses two key challenges: the trivialization of opinions resulting from excessive summarization and the diversion of focus caused by distractor concepts introduced from images. These challenges stem from the inductive (bottom-up) nature of existing debating schemes. To address the issue, we propose a deductive (top-down) debating approach called Blueprint Debate on Graphs (BDoG). In BDoG, debates are confined to a blueprint graph to prevent opinion trivialization through world-level summarization. Moreover, by storing evidence in branches within the graph, BDoG mitigates distractions caused by frequent but irrelevant concepts. Extensive experiments validate that BDoG is able to achieve state-of-the-art results in ScienceQA and MMBench with significant improvements over previous methods. The source code can be accessed at https://github.com/open_upon_acceptance.

## CCS CONCEPTS

• **Computing methodologies** → **Multi-agent systems**.

## KEYWORDS

multi-modal reasoning; multi-agent debate; large language models

## 1 INTRODUCTION

Multimodal reasoning depends on two key aspects: the creation of a unified representation of semantics from different modalities and the integration of these diverse semantics while ensuring logical consistency. While the advancement in large language models (LLMs) has made it possible to represent the semantics in natural languages [1, 31], the integration of diverse semantics remains a challenging issue, even in exclusive NLP tasks. One approach to tackle this challenge is multi-agent debate (MAD), where multiple LLMs act as agents, each contributing their own perspectives on the target topic and reaching a consensus through debates [4, 19]. This scheme could be adopted by incorporating a specific LLM for each modality as an agent.

While being relatively unexplored in the multimodal domain, MAD encounters numerous challenges in a broader context. It may suffer from the *trivialization* of opinions, resulting from the summarization step performed at the conclusion of each debating round.

Permission to make digital or hard copies of all or part of this work for personal or classroom use is granted without fee provided that copies are not made or distributed for profit or commercial advantage and that copies bear this notice and the full citation on the first page. Copyrights for components of this work owned by others than the author(s) must be honored. Abstracting with credit is permitted. To copy otherwise, or republish, to post on servers or to redistribute to lists, requires prior specific permission and/or a fee. Request permissions from permissions@acm.org.
*ACM MM, 2024, Melbourne, Australia*
© 2024 Copyright held by the owner/author(s). Publication rights licensed to ACM.
ACM ISBN 978-x-xxxx-xxxx-x/YY/MM
https://doi.org/10.1145/nnnnnnn.nnnnnnn

The objective of this step is to seek agreement among the participating agents regarding their opinions. Consequently, this process can lead to the debate's focus being directed towards a general concept, serving as an adaptation to accommodate the diverse range of semantics. One example can be observed in the reasoning of the Multimodal Large Language Model (MLLM) depicted in Figure 1, where the image modality presents a diverse range of semantics, including *bear sedge, earthworm, collared lemming*, and others. As a consequence, this can result in the context and summary being trivialized, shifting the emphasis from *lichen* to a more generalized concept of the tundra ecosystem, wherein both *bilberry* and *mushroom* exhibit a high degree of correlation. Similar issue exists when MAD is employed, where the summarizer concludes the diverse semantics into general words like *ecosystem* and *food web*, making the conclusion less specific. In addition, MAD may encounter the issue of *focus diversion*, which occurs when Chain-of-Thoughts (CoT) is utilized and new concepts introduced are highly correlated with a particular concepts (*e.g.*, *mathematical model* [6]), leading to an increased weighting of that concept within the context.

We argue that these challenges arise due to the inductive nature of existing debating schemes, wherein agent opinions are gathered from disparate concepts at word-level and consensus is achieved through bottom-up summarization. This approach may be effective in confined NLP tasks [10, 11], where the topic is often limited to a small number of concepts and the application of CoT remains constrained. However, in a multimodal scenario, certain modalities (*e.g.*, images) are information-rich and have a higher likelihood of introducing distracting concepts [23]. Consequently, it increases the semantic divergence within the context and the likelihood of trivialization. The semantic divergence increases further when the impacts of those concepts are amplified through CoT, particularly when the newly introduced concepts exhibit biases towards certain concepts, resulting in focus diversion.

To address these issues, we propose an deductive reasoning scheme called Blueprint Debate on Graph (BDoG, *pronounced bee-dog*). BDoG is inspired by the blueprint debate that has been employed in real-world debates, which distinguishes itself from other debates by its concentration on evaluating and refining a proposal (*e.g.*, blueprint) to address specific challenges or issues. BDoG begins by aggregating concepts from modalities and incorporating with their relationships into an initial graph. This graph serves as a blueprint that confines the scope of the discussion rather than having it open to irrelevant semantics as in existing schemes. More importantly, BDoG conducts the debate in a top-down manner by marking down conclusions on the graph. This prevents trivialization as specific concepts are preserved rather than merged into general ones. This can be found from the example shown in Figure 1, where the scope is limited to the tundra ecosystem while specific concepts like *mushroom* and *lichen* are retained. Furthermore, the graph provide a compact and high-level guidance for the discussion

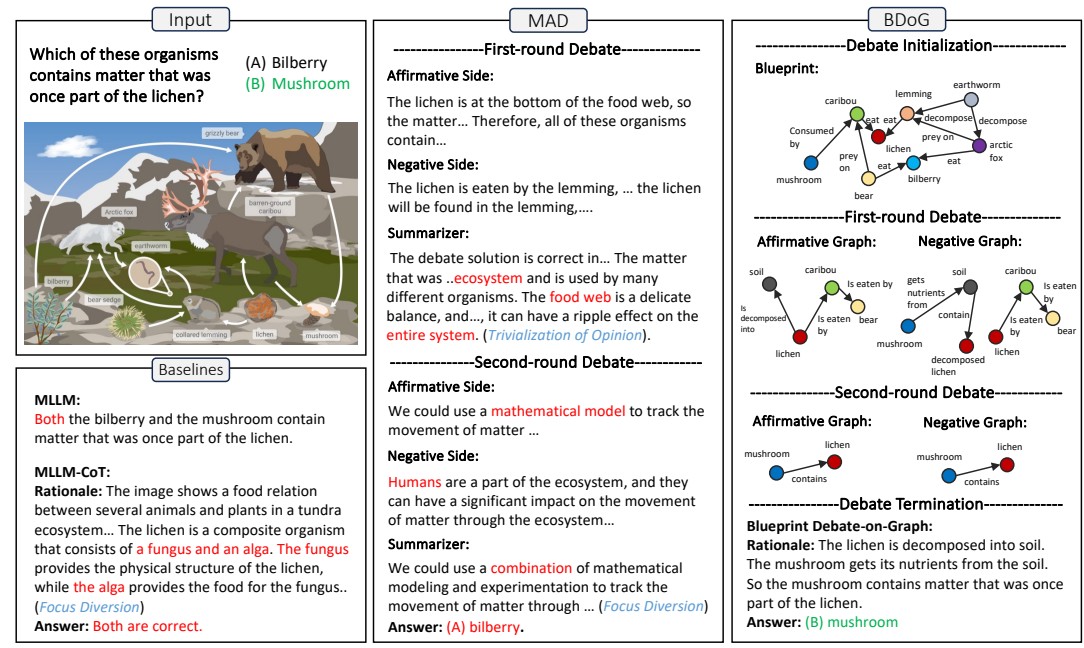

**Figure 1: Comparison results from ScienceQA dataset of direct answer from MLLM, Multimodal Chain-of-Thought (CoT), Multi-agent Debate (MAD) and our Blueprint Debate on Graph (BDoG). BDoG confines debates to a blueprint and stores evidence in graph branches, which mitigates word-level opinion trivialization and distractions caused by irrelevant concepts.**

process. The newly introduced concepts are incorporated into relevant branches instead of remaining as a word-level thoughts within the context. This reduces the likelihood of focus diversion since, in BDoG, the competition of semantics occurs at the branch level rather than the word level. This can be seen from Figure 1, where the most relevant branches related to the *soil* and *caribou* standout from the competition, eliminating the irrelevant semantics effectively. In addition to the advantages of scope-confined guidance and branch-level competition, BDoG also increases explainability, allowing for the tracking of discussion progress (Figure 1).

## 2 RELATED WORKS

### 2.1 Multimodal Reasoning

Multimodal reasoning is a crucial component in the development of advanced artificial intelligence (AI) systems that aim to replicate human-like intelligence [23]. This type of reasoning enables AI systems to process and analyze information from various sources and forms, such as text, images, audio, and video, in a integrated and coordinated manner [3, 28]. The latest advancements in multimodal large language models, such as BLIP2 [16], KOSMOS [15] and LLaVA [21] have demonstrated significant progress in complex reasoning, as these models [42] now have the capability to generate step-by-step rationales prior to producing the final answer, following a chain-of-thought (CoT) manner. Zheng et al. [46] propose a duty-distinct prompting method wherein questions are decomposed into sub-questions to enable deep-layer reasoning. SCITUNE [12] and T-SciQ [33] aim to teach large language models to answer science questions via the generation of mixed rationales derived from both large pretrained models and human annotators. Chameleon [24]

accomplishes complex multimodal reasoning tasks by integrating various external tools (*e.g.*, large language models, off-the-shelf vision models, and web search engines). Nevertheless, existing methods exhibit limitations as they heavily rely on either few-shot learning or supervision to "guide" the reasoning process. In order to overcome this dependency, we propose the incorporation of debating features into our method. This enables the agents to engage in an adversarial discussion, allowing them to "figure out" the correct direction autonomously. As a result, our approach makes zero-shot learning feasible by reducing the reliance on external guidance or supervision.

### 2.2 Multi-agent Debate

To mitigate the susceptible error in CoT reasoning, Shinn et al. [29] and Madaan et al. [25] employ model to reflect on task feedback signals that can induce better decision-making in subsequent trials. [45] exploit previously generated answer as hint to progressively guide towards correct answer. Although these methods effectively enhance the performance of LLM, they struggle to produce novel ideas once they have determined a response, as they rely solely on internal representations for generation [14]. Researchers are currently developing multi-agent collaborative systems to address above issues in pure-textual scenarios [40]. By designing these systems, large language models (LLMs) can work together to complete tasks or engage in productive debates by offering contrasting perspectives [4, 8, 19]. Zhang et al. [41] further reveal the collaboration mechanism from a social psychology view. This paper represents an initial endeavor to expand upon this method to facilitate multimodal reasoning. By incorporating multiple perspectives from

different multimodal language models, we can help address some of the limitations of individual models. Moreover, we address the trivialization of opinions and focus diversion problems of vanilla multi-agent debate via Blueprint Debate on Graph (BDoG).

## 2.3 Graph-augmented LLMs

Prior research has investigated the integration of structured graphs, such as knowledge graphs (KGs), into large language models (LLMs) by embedding the knowledge into the underlying neural networks [20, 35]. Nevertheless, embedding KGs within LLMs may compromise the inherent explainability and adaptability associated with knowledge reasoning and updating [13]. To tackle these challenges, recent studies have put forth innovative solutions. Li et al. [18] propose an adaptive query generator, facilitating the creation of queries across various query languages (*e.g.*, SPARQL) to infer rationales. Wang et al. [32] devise a structured multi-round question-answering (QA) format, which extracts external knowledge and generates coherent reasoning traces grounded in precise answers. Sun et al. [30] introduce Think-on-Graph (ToG), a method that sequentially reasons over KGs to locate relevant triples, thereby supporting the LLM in predicting the final answer. In the context of multimodal reasoning, CCoT [26] substitutes the rationale generation process with scene graph extraction to enhance the compositional capabilities of large multimodal models. KAM-CoT [27], on the other hand, incorporates external KGs during the two-stage training process, yielding state-of-the-art fine-tuning outcomes in multimodal reasoning. In contrast to existing methods that utilize static graphs, our proposed BDoG preserves the dynamics and precision of KGs through iterative updates of entities, attributes, and relationships, guided by a blueprint debate process.

## 3 PRELIMINARY

We begin by outlining existing approaches for tackling the multimodal reasoning problem. Figure 2 shows the specific distinction among them. Formally, given a question $Q$ consisting of $t$ tokens, our goal is to identify the correct answer $A$ from a set of candidate answers. In the context of multimodal reasoning, the expected answer is intended to be inferred based on a visual context $I$ and a textual clue $C$, in addition to the question itself.

**Vanilla Prompting.** Vanilla prompting approaches aim to predict an answer $A$ by augmenting the input with illustrative examples $D$ in addition to the question $Q$, visual context $I$, and textual clue $C$.

**Multimodal CoT.** As noted by Lu et al. [23], incorporating intermediate reasoning steps (rationales) can aid in predicting the correct answer, especially for complex multimodal reasoning tasks. To address this, we first generate a rationale $R = \{r_1, r_2, ..., r_k\}$ given the input. The generated rationale $R$ is then concatenated with the original language input to update the language representation. This augmented language input is fed together with the original visual input $I$ into the same model to infer the final answer.

**DDCoT.** The Duty-Distinct Chain of Thought framework proposes a novel approach for deconstructing questions into fundamental sub-questions, similar to breaking down reasoning into elementary steps. Contrary to prior work on conversational agents, Zheng et al. [46] employ the instruction to acquire the sub-question sequence $Q_1, Q_2, ..., Q_t$ in a single interaction. Within this framework, the final response $A$ is obtained by aggregating the answers $A_i$ to each sub-question $Q_i$ and the generated CoT rationale $R_i$.

**Self-Correction.** Self-correction techniques [37] endeavor to iteratively enhance model predictions by leveraging feedback generated from the model itself. In particular, a *feedback* function $f : R \rightarrow R'$ is adopted to iteratively map model outputs to the refined responses.

**MAD.** MAD [19] presents a promising framework that fosters discursive exchange and cross-pollination of ideas between conversational models. Consider a debate comprising $j$ rounds amongst a set of large language models acting as interlocutors, the *proponent* generates a rationale $R'_p$ and response $A_p$ revised in the light of rationales $R_o$ presented by the *opponent* in prior turns.

## 4 BLUEPRINT DEBATE ON GRAPH

In this section, we introduce Blueprint Debate-on-Graph (BDoG). As illustrated in Figure 2, BDoG takes a deductive approach instead of inducing answers from word-level thoughts. It utilizes graphs to structure the opinions and proposals provided by agents. This graph-level structuring of the debating context helps to minimize opinion trivialization and focus diversion. Furthermore, BDoG adopts a top-down approach which improves multimodal reasoning by iteratively refining an initial proposal, represented as a blueprint graph. This integrates opinions from diverse perspectives through the competition and cooperation among multiple agents.

The BDoG at the $i^{th}$ round can be formulated as a quadruple

$$\mathcal{T}^i = (\mathcal{G}^i, \mathcal{S}, \mathcal{A}, \mathcal{F}) \tag{1}$$

where, given a multimodal source set $\mathcal{S} = \{Q, I, C\}$, the debating is conducted among a set of agents $\mathcal{A} = \{a_j\}, j \in \mathbb{Z}^+$, in which each agent uses operations from the set $\mathcal{F} = \{f_k\}, k \in \mathbb{Z}^+$ to propose opinions by refining the graph-of-thoughts $\mathcal{G}^i$. At the end of the $i^{th}$ round, $\mathcal{G}^i$ is updated to $\mathcal{G}^{i+1}$ to initiate the next round.

## 4.1 Blueprint $\mathcal{G}^0$ Initialization

To initiate the debating, we need to convert the multimodal sources into a blueprint graph. This conversion is achieved through the operation function $f_0 \in \mathcal{F} : \mathcal{S} \mapsto \mathcal{G}^0$. To implement $f_0$, we define two additional sub-functions $f_t$ and $f_v$ for extracting entities and relations from the textual sources (*i.e.*, $Q$ and $C$) and visual source (*i.e.*, $I$), respectively. The implementation of $f_0$ is formulated as

$$f_0 : \mathcal{S} \mapsto \mathcal{G}^0$$
$$f_t(Q) \cup f_v(I) \cup f_t(C) \mapsto \langle \mathcal{V}^0, \mathcal{E}^0 \rangle$$
$$w.r.t \quad Size, \; Relevance \tag{2}$$

where $\cup$ denotes the union of two sets of graphs. The 2 constraints are as follows: 1) **Size Constraint**: The size of $\mathcal{G}^0$ needs to be restricted within a specific range to prevent an excessive number of clues that could distract the inference or an insufficient number to answer the question effectively. 2) **Relevance Constraint**: We should merge the relationships extracted from $I$ and $C$ towards those of the question $Q$, ensuring all the knowledge encapsulated in $\mathcal{G}^0$ is relevant to the question. Extensive libraries are available for $f_t$ and $f_v$, as they have been extensively researched (*e.g.*, named entity recognition [38], relation extraction [44] for $f_t$, image captioning [43], visual grounding [17] for $f_v$). However, the recent advancements in multimodal large language models (MLLM) have made

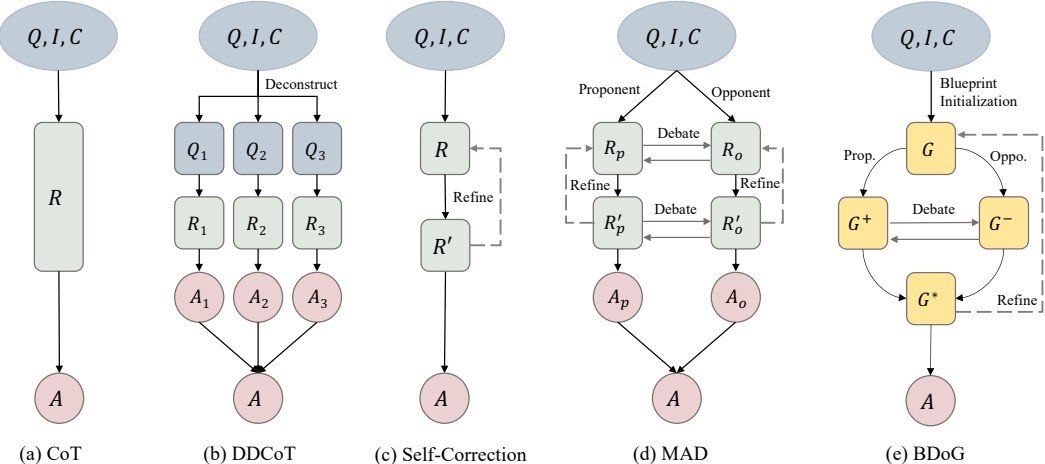

**Figure 2: Comparison of CoT, Duty-Distinct CoT (DDCoT), Self-Correction, Multi-agent Debate (MAD) and Our proposed Blueprint Debate on Graph (BDoG). Q: input question, I: input image, C: context or hint, A: answer, R: rationale, G: blueprint.**

it convenient to implement these sub-functions using in-context learning based prompts. For example, to extend the query $I$ in the context, we can employ CoT to implement $f_t$ as

> $f_t(Q)$: *Given the question {Q}, please provide the necessary steps to answer this question.*

where the {} denotes the placeholder in the prompt.

For $f_v(I)$, its implementation varies depending on LLMs used. For GPT-4, the image needs to be encoded in Base64 format. Gemini utilizes PIL for image encoding. InstructBLIP offers its EVA-G encoder to convert the image into an eigenvector. The $f_0$ can then be implemented as

> $f_0$: *Given the image {$f_v(I)$} and question {$f_t(Q)$}, generate a scene graph with evidence to answer the question. Please ensure adherence to following constrains: {Size}, {Relevance}.*

where two exemplar constraints are

> *Size* : *The graph must not be empty. Please restrict the maximum number of objects in the graph to 20.*
>
> *Relevance* : *The objects and relations within the graph should be pertinent to addressing the question.*

It worth mentioning that although we provide some exemplar implementations of functions and constraints, the effectiveness of prompts can vary significantly depending on the MLLM used. The success of multimodal reasoning relies more on the development of guiding principles for prompting the models and constraints for regularizing the resulting graph. Therefore, in the rest of this section, our focus lies on discussing these guiding principles and constraints. Our prompt implementations will be provided in Appendix.

## 4.2 Agents and Roles

In the debate, we can treat each LLM as an agent that participates in the discussion by refining the blueprint graph $\mathcal{G}^0$. Just like in a real debate, each agent $a_j \in \mathcal{A}$ has a distinct role assigned. We define three roles as a set of $\mathcal{R} = \{Proponent, Opponent, Moderator\}$.

These roles not only help structure the discussion but also promote critical thinking and ensure a comprehensive and in-depth exploration of the topic.

**Proponent** agents advocate and defend the current blueprint by refining current $\mathcal{G}^i$ into an affirmative evidence graph $\mathcal{G}^+$. A debating function is assigned for this purpose as

$$\begin{aligned} Proponent\ f^+ :\ &\mathcal{G}^i \times \mathcal{S} \mapsto \mathcal{G}^+ \\ &\langle \mathcal{V}^i, \mathcal{E}^i \rangle \cup f_t(Q) \cup f_v(I) \mapsto \langle \mathcal{V}^+, \mathcal{E}^+ \rangle \\ &w.r.t\ \ Size,\ Relevance,\ Compactness \end{aligned} \tag{3}$$

An exemplar implementation is

> $f_+$: *As {personality}, you are assigned as an **affirmative debater** and have been provided with an evidence graph {$\mathcal{G}^i$} for answering the question {$f_t(Q)$} related to the image {$f_v(I)$}. Try to enhance the graph by incorporating your insights towards an optimal solution. Please ensure adherence to following constrains: {Size}, {Relevance}, {Compactness}.*

Note that we have incorporated the conclusion from [8, 40] that the agent's understanding of the role can be improved by using the {*personality*} for targeted personality injection. Furthermore, the personality can be tailored to be specific, such as "Ben, a high school student with an impressive academic record and respected by peers for your knowledge and logical thinking." The Proponent debate adheres to the Size and Relevance constraints defined in Eq. (3), and it also includes the **Compactness Constraint**: The refined graph should be as concise as possible, ensuring that the blueprint remains focused.

**Opponent** agents challenge and present arguments against the blueprint $\mathcal{G}^+$ by updating it into a negative evidence graph $\mathcal{G}^-$ as

$$\begin{aligned} Opponent\ f^- :\ &\mathcal{G}^+ \times \mathcal{S} \mapsto \mathcal{G}^- \\ &\langle \mathcal{V}^+, \mathcal{E}^+ \rangle \cup f_t(Q) \cup f_v(I) \mapsto \langle \mathcal{V}^-, \mathcal{E}^- \rangle \\ &w.r.t\ \ Size,\ Relevance,\ Compactness \end{aligned} \tag{4}$$

An exemplar implementation is

> $f_+$: As {personality}, you are assigned as a **negative debater** and have been provided with an affirmative evidence graph {$\mathcal{G}^+$} for answering the question {$f_t(Q)$} regarding the image {$f_v(I)$}. Try to detect potential flaws and drawbacks of the graph and update it with your insights. Please ensure adherence to following constrains: {Size}, {Relevance}, {Compactness}.

The utilization of the functions $f_+$ and $f_-$ fosters an adversarial dynamic between the Proponent and Opponent, ensuring a diverse and comprehensive discussion.

To facilitate the debating, **Moderator** agents synthesize the arguments and opinions presented by both the proponent and opponent by merging the $\mathcal{G}^+$ and $\mathcal{G}^-$ into a conclusion $\mathcal{G}^*$ as

$$\text{Moderator } f_*: \mathcal{G}^+ \cup \mathcal{G}^- \mapsto \mathcal{G}^*$$
$$\langle \mathcal{V}^+, \mathcal{E}^+ \rangle \cup \langle \mathcal{V}^-, \mathcal{E}^- \rangle \mapsto \langle \mathcal{V}^*, \mathcal{E}^* \rangle$$
$$w.r.t \ Size, \ Relevance, \ Compactness \quad (5)$$

An exemplar implementation is

> $f_*$: As {personality}, you are assigned as a **moderator** in a debate and have been provided with an affirmative evidence graph {$\mathcal{G}^+$} and a negative evidence graph {$\mathcal{G}^-$} to address the question {$f_t(Q)$} regarding the image {$f_v(I)$}. Try to consolidate the two graphs into a single graph towards the optimal solution, and provide a conclusive answer to the question.

### 4.3 Debate Progress and Graph Condensation

**Initialization and Role Assignment**: Once the blueprint $\mathcal{G}^0$ has been initialized, the debate commences with the assignment of roles to agents in $\mathcal{A}$. Denote the assignment of a role $r \in \mathcal{R}$ to an agent $a_j$ as $a_j := r$, to ensure a balanced debate, an equal number of agents are assigned as Proponents and Opponents, with only one agent assigned as the Moderator. The *Role Assignment Regulation* is

$$\|\{a_j | a_j := Proponent\}\| = \|\{a_k | a_k := Opponent\}\|,$$
$$\|\{a_l | a_l := Moderator\}\| = 1.$$

**Debating**: After roles are assigned, the debate can be conducted iteratively between the Proponents and Opponents as illustrated in Figure 2. The initial blueprint $\mathcal{G}^0$ is then updated in subsequent debate rounds. In each round, the Moderator summarizes the affirmative and negative graphs in a conclusion graph on the basis of which a tentative answer is also provided. If the debate is not concluded, the Moderator initiates the next round by assign $\mathcal{G}^-$ as the blueprint $\mathcal{G}^{i+1}$. Otherwise, the Moderator's answer is considered final and adopted.

**Stopping Criteria**: The condition to conclude the debate can be determined by assessing the modifications made to the evidence graph compared to the previous round as

$$\|\mathcal{G}^{i+1} - \mathcal{G}^i\| \leq \epsilon \quad (6)$$

where $\| \cdot \|$ is a distance metric defined on the graphs. The rationale is that with each successful round of debate, the evidence becomes more concise, leading to the condensation of the evidence graph. Therefore, we can quantify the modification by tallying the number

of entities (relations) that have been updated and pruned as

$$\|\mathcal{G}^{i+1} - \mathcal{G}^i\| = \|\langle \mathcal{V}^{i+1}, \mathcal{E}^{i+1} \rangle - \langle \mathcal{V}^i, \mathcal{E}^i \rangle\|$$
$$= \|\{\mathcal{V}^{i+1} \cap \mathcal{V}^i\}\| + \|\{\mathcal{E}^{i+1} \cap \mathcal{E}^i\}\|$$
$$+ \|\{\mathcal{V}^i - \mathcal{V}^{i+1} \cap \mathcal{V}^i\}\| + \|\{\mathcal{E}^i - \mathcal{E}^{i+1} \cap \mathcal{E}^i\}\|. \quad (7)$$

## 5 EXPERIMENTS

### 5.1 Backbone Models

To evaluate its performance and generalizability, we have implemented Blueprint Debate-on-Graph (BDoG) using different prevalent multimodal large language models as backbones, including 1) **GeminiProVision** [31], an extensively parameterized model developed by Google, 2) **InstructBLIP** [7] and **LLaVA-v1.5** [21], which possesses more constrained dimensions and computational resources relative to alternative architectures, and 3) **GPT-4** [1] which is the fourth iteration of the GPT model developed by OpenAI. More implementation details can be found from the Appendix.

### 5.2 Datasets and Metrics

In line with the general setup described in [24, 46], we perform our experiments using two extensively adopted multimodal question answering (QA) datasets. These datasets are widely recognized as standard benchmarks, specifically designed to evaluate the performance and effectiveness of models in addressing multimodal reasoning tasks. The two benchmarks are: 1) **ScienceQA-IMG** (SQA-IMG) [23] represents the first multimodal scientific question-answering corpus comprising 21,000 inquiries paired with multiple choices and accompanying images. As a *training-free* approach, we solely utilize the TEST and DEV partitions of ScienceQA-IMG following prior work [23] for comparative assessment. 2) **MMbench** [22] offers a more systematic and robust means for *zero-shot* reasoning evaluation compared to existing benchmarks such as VQAv2 [9] or COCO Captions [5]. We employ the official data split (MMBench-Dev) and code released by the originating authors. We report the accuracy metric through a heuristic matching procedure, following the same setting of the official benchmark [23]. The statistics of the two benchmarks and detailed settings are delineated in Appendix.

### 5.3 Performance Comparison to SOTA Methods

In contrast to the few-shot methodology, which exhibits susceptibility to the specific examples selected for training, we have opted for the zero-shot setting. This approach circumvents potential biases introduced by a limited sample size, ensuring a more robust and generalizable model. We evaluate the proposed method by by comparing it against two sets of SOTA approaches as follows:

- Open-Source Multimodal LLMs with Relatively Moderate Parameters including MiniGPT-4 [47], Qwen-VL and Qwen-VL-Chat [2], CogVLM-Chat [34], mPLUG-Owl2 [39], LLaVA-v1.5 [21], and InstructBLIP [7], with parameter scales ranging from 7B to 17B.
- Closed-Source Multimodal LLMs with Large-Scale Parameters: GPT-3.5 [36], GPT-4V [1] and GeminiProVision [31]. Following the general standard, GPT-3.5 and GPT-4 have been incorporated with the CoT [36] or DDCoT [46] (built based on image captioning results). These models are known for their parameter scales above 175B and are considered to have the best performance.

| Model | Size | SQA-IMG | MMBench |
|---|---|---|---|
| MiniGPT-4 [47] | 7B | 37.7 | 24.3 |
| Qwen-VL [2] | 7B | 58.6 (67.1) | 38.2 |
| Qwen-VL-Chat [2] | 7B | 68.6 (68.2) | 60.6 |
| mPLUG-Owl2 [39] | 8B | 63.9 | 66.5 |
| CogVLM-Chat [34] | 17B | 69.6 | 63.7 |
| InstructBLIP [7] | 13B | 59.2 (63.1) | 44.0 |
| InstructBLIP+**BDoG** | 13B | **63.5** | **55.8** |
| LLaVA-v1.5 [21] | 13B | 71.6 | 68.2 |
| LLaVA-v1.5+**BDoG** | 13B | **72.0** | **71.1** |
| GPT-3.5+CoT [36] | 175B | 67.4 | - |
| GPT-3.5+DDCoT [46] | 175B | 72.5 | - |
| GPT-4+CoT [36] | 175B+ | 71.5 | 75.1 |
| GPT-4+**BDoG** | 175B+ | **77.2** | **79.2** |
| GeminiProVision [31] | 175B+ | 76.5 | 75.2 |
| GeminiProVision+**BDoG** | 175B+ | **81.1** | **81.3** |

**Table 1: Overall zero-shot results on ScienceQA-IMG test set and MMBench dev set. Size = backbone model size. There are limited zero-shot results previously published on ScienceQA-IMG, so we reimplemented above models and report our findings. Where possible, we include results from the LLaVA paper for comparison (shown in parentheses). For MMBench, we refer to the scores listed on the official public leaderboard.**

The results are shown in Table 1. The integration of BDoG has resulted in a significant improvement across different backbones, as evidenced by the performance gains of 4.3% ∼ 5.7% on SQA-IMG and 6.1% ∼ 11.8% on MMBench. Notably, when combined with GeminiProVision, BDoG achieves SOTA performance on the ScienceQA-IMG test set and MMBench development set, achieving accuracies of 81.1% and 81.3%, respectively. Other observations that indicate BDoG's advantage over SOTA methods include:

**BDoG helps reduce the performance gap between large and small models.** It is commonly believed that models with larger parameter scales tend to perform better than smaller ones. This observation generally holds true, as shown in Table 1 for models without BDoG. However, the introduction of BDoG has led to a reduction in the performance gap between these two types of models. This can be seen in the improvement achieved by InstructBLIP, which has experienced a boost of 4.3% and achieves an accuracy of 63.5% on SQA-IMG, comparable to that of GPT-3.5. Similar results can be found in LLaVA-v1.5 with BDoG which gains the 71.1% accuracy in MMBench, comparable to the GPT-4 model.

**BDoG reinforces the multimodal reasoning.** Form Table 1, we can also observe the advantage of direct multimodal reasoning (*e.g.*, open-source VL models, and GeminiProVision) over indirect multimodal reasoning (*e.g.*, GPT3.5+CoT and GPT3.5+DDCoT due to their nature of obtaining visual information through image captioning). Even the open-source VL models of the former group achieves comparable performance to those of the latter one, with much smaller parameter scales. With BDoG, which reinforces multimodal reasoning by graph regulation, the performance of direct multimodal reasoning of InstructBLIP and GeminiProVision have been improved by 6.1% and 11.8% on the MMBench dataset.

## 5.4 Ablation Study

In order to gain a comprehensive understanding of BDoG, we conduct an ablation study by decomposing BDoG into two variants:

- **BDoG**$^{Debate}$: we remove the graph regulation and constraints, resulting in a debate-only approach (*i.e.*, vanilla multi-agent debate) for investigating the specific contribution of the debating component of BDoG.
- **BDoG**$^{Graph}$: we remove the debating rounds, resulting in a graph-based reasoning method for investigating the specific contribution of the graph regulation component of BDoG.

Moreover, we analyze the performance of the two variants on the benchmarks by breaking it down into subcategories. This analysis allows us to investigate the preferences of these two variants for different types of questions. The results are presented in Table 2, where it can be observed that both variants demonstrate comparable performance across various benchmarks. This suggests that the debate and graph components of BDoG contribute to its effectiveness in a similar manner. Through the combination of these two components in BDoG, the performance has experienced further improvement compared to the individual variants. However, when considering specific categories, distinctions in the contributions of the debate and graph components become apparent.

**Impact of the debate component: BDoG**$^{Debate}$ demonstrates consistent improvements across both benchmarks with a debate-only setting, which encourages LLM agents to collaboratively refine and correct prior responses. For science questions, **BDoG**$^{Debate}$ facilitates the model's focus on specific errors, such as direction, size, and position, leading to improved performance in the natural science domain (boosting accuracy from 53.7 to 59.7 for InstructBLIP and 68.9 to 73.3 for GeminiProVision). However, the debate-only nature has limitations, including *trivialization* and *focus diversion* issues. Without the graph regulation, overall performance decreases from 55.8 to 52.4 for InstructBLIP, particularly when addressing questions that require attention to multi-hop logistic reasoning (LR) and specific attributes (AR).

**Impact of the graph regulation:** With a graph-regularized knowledge base for the discussion, **BDoG**$^{Graph}$ also demonstrates consistent improvement of 2.3% ∼ 7.1% overs the base models on both benchmarks. Compared to the text-based and debate-only method **BDoG**$^{Debate}$, it performs evidently better on the logistic reasoning and attributes reasoning questions by addressing the opinion trivialization and diversion with initialized blueprint. Although incorporating fact-related graph information proves beneficial in **BDoG**$^{Graph}$, the absence of the iteratively refined debate procedure results in decreased performance due to the coarse and distorted extraction of blueprint information.

**Impact of combining the debate and graph components:** By combining the two components, BDoG achieves gains across nearly all categories. In the ScienceQA-IMG dataset, BDoG exhibits consistent and steady improvements, averaging around 5% compared to the baseline models. This suggests that BDoG is robust and generalizes well for science-related questions. Remarkably, BDoG significantly outperforms the baseline model (InstructBLIP) on the MMBench-Dev set, particularly in the areas of Logical Reasoning (LR) with a margin of 44.2%, Attribute Reasoning (AR) with a margin of 17.7%, and Relation Reasoning (RR) with a margin of 3%. BDoG

| Model | Method | ScienceQA-IMG-Dev | | | | ScienceQA-IMG-Test | | | | MMBench-Dev | | | | | | |
|---|---|---|---|---|---|---|---|---|---|---|---|---|---|---|---|---|
| | | NAT | SOC | LAN | Avg | NAT | SOC | LAN | Avg | LR | AR | RR | FP-S | FP-C | CP | Avg |
| MniGPT-4 [47] | | 42.9 | 30.6 | 43.7 | 38.4 | 42.0 | 30.1 | 50.0 | 37.7 | 7.5 | 31.3 | 4.3 | 30.3 | 9.0 | 35.6 | 24.3 |
| Qwen-VL [2] | | 52.1 | 59.8 | 58.3 | 55.0 | 55.7 | 62.0 | 77.3 | 58.7 | 16.1 | 44.7 | 34.8 | 35.2 | 39.2 | 46.6 | 38.2 |
| Qwen-VL-Chat [2] | Base | 60.9 | 67.4 | 62.5 | 63.3 | 67.7 | 69.6 | 75.0 | 68.6 | 32.2 | 59.8 | 43.5 | 66.2 | 48.3 | 79.4 | 60.6 |
| mPLUG-Owl2 [39] | | 60.6 | 68.0 | 45.8 | 62.8 | 62.5 | 66.2 | 61.4 | 63.9 | 32.2 | 72.4 | 60.9 | 68.6 | 60.1 | 79.4 | 66.5 |
| CogVLM-Chat [34] | | 63.1 | 69.2 | 77.1 | 65.6 | 68.0 | 72.2 | 70.4 | 69.7 | 29.7 | 65.8 | 60 | 66.9 | 58 | 76.7 | 63.7 |
| LLaVA-v1.5 [21] | | 66.1 | 74.9 | 72.9 | 69.4 | 70.1 | 74.2 | 81.8 | 71.9 | 44.1 | 67.3 | 60.0 | 72.0 | 59.4 | 82.1 | 68.2 |
| InstructBLIP [7] | Base | 53.7 | 57.3 | 47.9 | 54.8 | 58.1 | 61.0 | 61.4 | 59.2 | 19.1 | 54.2 | 34.8 | 47.8 | 24.8 | 56.4 | 44.0 |
| | + BDoG$^{Debate}$ | 59.7 | 55.6 | **54.2** | 58.1 | **63.1** | 58.2 | 72.7 | 61.4 | 22.9 | 60.3 | **52.2** | 54.3 | **28.0** | 68.9 | 52.4 |
| | + BDoG$^{Graph}$ | 58.1 | 61.3 | 52.1 | 59.0 | 60.6 | 62.6 | 68.2 | 61.5 | 58.8 | 65.5 | 41.2 | 51.2 | 18.6 | 46.1 | 51.1 |
| | + BDoG | **61.1** | **64.0** | 52.1 | **61.9** | 61.1 | **66.5** | **75.0** | **63.5** | **63.3** | **71.9** | 37.8 | **56.3** | 20.3 | 59.1 | **55.8** |
| GeminiProVision [31] | Base | 68.9 | 81.6 | 75.0 | 73.7 | 72.9 | 81.5 | 88.6 | 76.5 | 55.9 | 80.4 | 73.9 | 79.5 | 61.5 | 82.1 | 75.2 |
| | + BDoG$^{Debate}$ | 73.3 | 81.1 | 77.1 | 76.2 | 75.3 | 82.8 | **93.2** | 78.5 | 71.1 | **85.1** | 83.1 | 78.9 | 71.9 | 81.3 | 79.3 |
| | + BDoG$^{Graph}$ | 69.8 | 84.8 | **87.5** | 75.6 | 74.7 | 86.8 | 88.6 | 79.6 | **75.0** | 84.5 | 80.7 | **81.4** | 73.0 | 83.6 | 80.7 |
| | + BDoG | **73.6** | **86.2** | 85.4 | **78.4** | **76.6** | **87.4** | 93.2 | **81.1** | 74.0 | 84.8 | **83.4** | 81.3 | **73.7** | **84.4** | **81.3** |

**Table 2: Ablation study on ScienceQA-IMG dev and test set and MMBench dev set. Question classes: NAT = natural science, SOC = social science, LAN = language science, LR = Logical Reasoning; AR = Attribute Reasoning; RR = Relation Reasoning; FP-S = Fine-grained Perception (Single Instance); FP-C = Fine-grained Perception (Cross Instance); CP = Coarse Perception.**

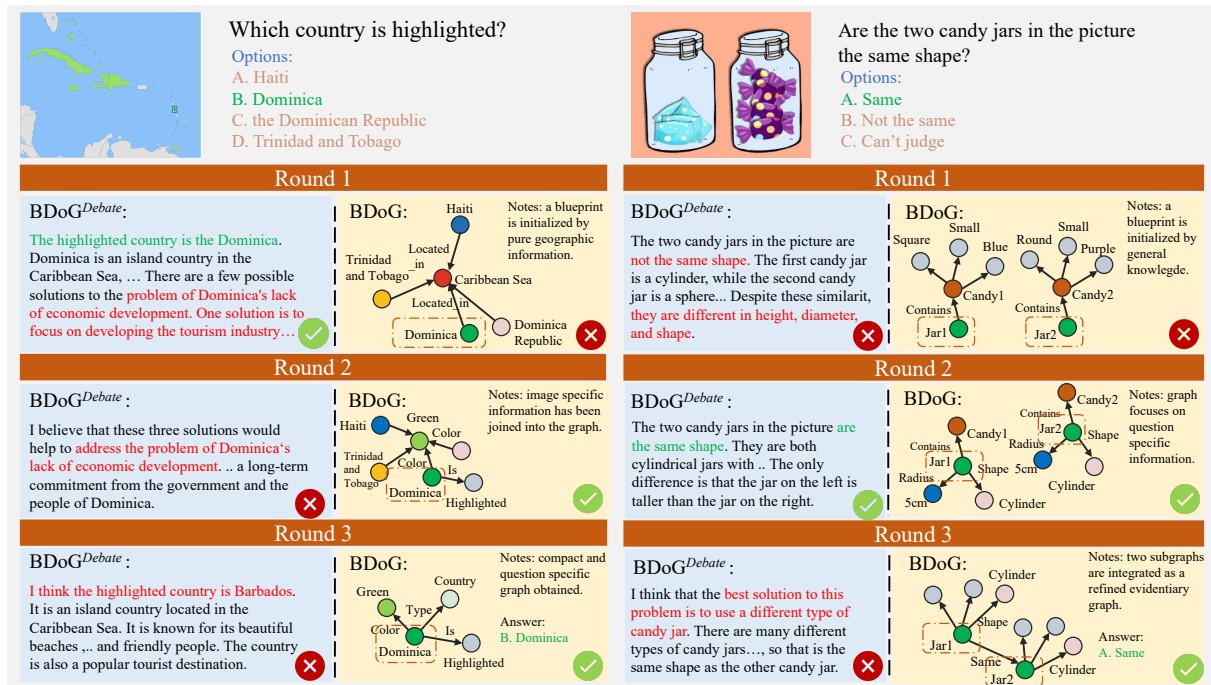

**Figure 3: Case study of our proposed Blueprint Debate on Graph (BDoG) and vallina Multi-agent Debate (BDoG$^{Debate}$) on ScienceQA-IMG (left) and MMBench (right) datasets. Green color indicates the correct answer/rationale and Red means incorrect/irrelevant predictions.**

enhances logical reasoning (LR) through a mechanism that refines the reasoning process iteratively, emphasizing the importance of multi-step reasoning rationales. The blueprint graph structure of BDoG, which explicitly models objects, attributes, and relations, contributes to improved reasoning abilities in Attribute Reasoning (AR) and Relation Reasoning (RR). The GeminiProVision model also exhibits comparable performance improvements, with BoG contributing to enhanced fine-grained perception across instances (FP-C), resulting in a gain of 12.2%. This improvement can be attributed to the connections established between various objects within the debate-on-graph framework.

**A case study for the iterative improvement on the blueprint:** BDoG leverages the advantages of both structured evidence through graph regulation and iterative refinement through debating. This is evident in the consistent improvement observed on the blueprint graph, showcasing the combined benefits of these two components. Figure 3 provides running examples demonstrating the superior reasoning performance of our proposed BDoG framework compared to the **BDoG$^{Debate}$** method.

The left case draws from the ScienceQA dataset, testing geographic knowledge and map interpretation. While **BDoG$^{Debate}$** correctly answered *Dominica is highlighted*, it also generated irrelevant information about Dominica's economic development. This

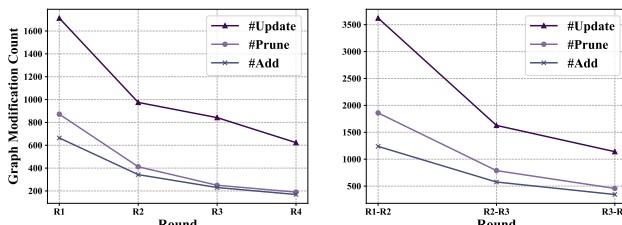

**Figure 4: Statistics of intra-round (left) and inter-round (right) Blueprint condensation of BDoG with GeminiProVision for ScienceQA-IMG test set. #Update: number of updated attributes; #Prune: number of pruned entities/relations; #Add: number of newly-added entities/relations.**

misguided the agents into off-topic discussion, concluding incorrectly with *Barbados*. In contrast, BDoG concentrated on the question and options, iteratively refining the blueprint entities and relations to arrive at the right answer of Dominica.

The example on the right comes from the MMBench dataset requiring cross-instance perception. As the image contained both *candies* and *jars*, it posed a challenge. With **BDoG**$^{Debate}$ relying on text alone, agreement was rarely reached as responses changed over debate rounds. However, BDoG first generated a blueprint defining image objects and attributes. This established the discussion scope. BDoG then pruned irrelevant *candy* information, focusing discussion on the specific object - *jars*. It output the final answer by comparing and connecting the two *jar* sub-graphs.

In summary, Figure 3 demonstrates that BDoG beats **BDoG**$^{Debate}$ on both datasets through its blueprint-driven approach. This concentrates graph-based reasoning on salient topics and prunes irrelevant details to arrive at well-supported conclusions.

| Round | ScienceQA-IMG-Test | | MMBench-Dev | |
|---|---|---|---|---|
| | BDoG-S | BDoG-L | BDoG-S | BDoG-L |
| 1 | 60.5 | 80.6 | 51.6 | 81.0 |
| 2 | **63.5** | 80.9 | 54.6 | 81.1 |
| 3 | 63.1 | 81.1 | **55.8** | **81.3** |
| 4 | 63.3 | **81.4** | **55.8** | 80.9 |

**Table 3: Model performance with respect to the iteration round of debate. BDoG-S: InstructBLIP with BDoG, BDoG-L: GeminiProVision with BDoG.**

## 5.5 Monitoring The Debating Progress

We evaluate the model's performance against the termination criteria across multiple debate rounds based on the data in Table 3. Our analysis shows that for models with smaller parameters like InstructBLIP, moving from a single round to two rounds led to significant gains in performance. This improvement is particularly notable when increasing the number of rounds from one to two. However, for larger models that may reach agreement more easily, the performance enhancement is relatively modest when amplifying the number of debate rounds. In general, we find the model's performance tend to converge within the second or third round.

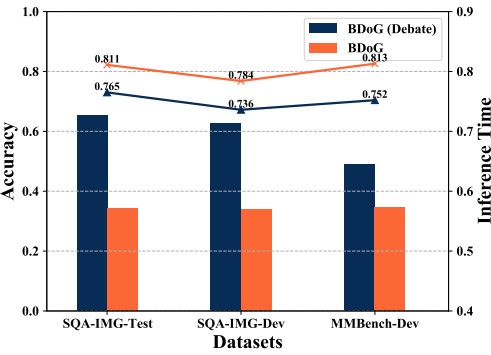

**Figure 5: Effectiveness vs. efficiency results, comparing our proposed Blueprint Debate-on-Graph (BDoG) and vanilla Multi-agent Debate (BDoG (Debate)) on GeminiProVision. The bar chart indicates the inference time on three datasets and lines indicate the zero-shot performance (Accuracy).**

This can be attributed to the underlying reasoning typically being able to answer questions within 2-3 steps.

Additionally, Figure 4 illustrates the number of updated attributes, newly added or removed entities or relations between and within rounds. A strength of our proposed BDoG framework is its ability to quantify the debate process by inspecting graph changes. This demonstrates the effectiveness of dynamically adjusting the initial graph based on the discussion. The results in Figure 4 are also consistent with our hypothesis that disagreements and errors can be decreased as the debate progresses.

## 5.6 Efficiency Analysis

We further compare the effectiveness versus efficiency of our BDoG framework against **BDoG**$^{Debate}$, as shown in Figure 5. Maintaining concise content focuses on key aspects, the graph structure of BDoG demonstrates superior efficiency, requiring approximately 50% less inference time than **BDoG**$^{Debate}$. By first generating a blueprint, BDoG defines the scope of the current state, thereby improving model efficiency by filtering irrelevant information. Concurrently, Figure 5 shows BDoG outperforms **BDoG**$^{Debate}$ in effectiveness, achieving over 5 percentage higher accuracy than **BDoG**$^{Debate}$ across three test sets. This enhanced effectiveness can be attributed to BDoG's concentrating on salient knowledge rather than generational textual content without guidance, as in **BDoG**$^{Debate}$.

## 6 CONCLUSION

This paper has presented a pioneering pilot study that introduces multi-agent debate into the realm of multimodal reasoning. We tackled two prominent challenges faced in this context: the issue of opinions being trivialized and focus diversion. By recognizing the limitations of existing debating schemes, we propose Blueprint Debate on Graphs (BDoG), which confines debates to a blueprint graph and stores evidence in graph branches, to address the challenges of word-level opinion trivialization and distraction caused by irrelevant concepts. Extensive experiments conducted in Science QA and MMBench validate the efficacy of BDoG, surpassing previous methods and establishing new state-of-the-art results.

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
