# OpenReview forum: "A Picture Is Worth a Graph: A Blueprint Debate Paradigm for Multimodal Reasoning"
_acmmm.org/ACMMM/2024/Conference — MM2024 Oral_

### Official Review · Reviewer_TCTh · 2024-05-20

**Rating:** 4
**Confidence:** 3

**Summary:**

This paper integrates Multi-agent Debate with Graph-augmented Large Language Models (LLMs) and introduces a new paradigm Blueprint Debate on Graphs (BDoG) to address the challenges of opinion trivialization and focus diversion in multimodal reasoning. Specifically, the authors convert multimodal sources into graphs by designing prompts that leverage multimodal LLMs to extract entities and relationships from textual and visual information. They also design three agents (Proponent, Opponent and  Moderator) to refine the graph through multiple rounds of debate, ensuring it remains as concise as possible. This method helps mitigate issues of opinion trivialization and focus diversion. Experimental results on ScienceQA-IMG and MMbench demonstrate the effectiveness of the proposed approach.

**Strengths:**

1. Explored the effectiveness of LLMs in multimodal reasoning. The carefully designed prompts limit the graph size and maintain a strong correlation with the problem. Proponent and Opponent agents optimize the entities and relationships in the graph from different perspectives, with Moderator agents summarizing the results. This new paradigm enhances the model's reasoning capabilities while alleviating issues of opinion trivialization and focus diversion.
2. The contribution is demonstrated through ablation studies by decomposing BDoG into its "Debate" and "Graph" components.

**Limitations:**

1. How does the quality of blueprints generated by different multimodal LLMs differ? If the blueprints generated by the multimodal LLM with the highest quality are used for inference on different models, what is the difference in effect? As shown in Figure 5 in the appendix, the final correctness is highly dependent on the quality of the initialized blueprint. Even after iterative optimization, low-quality blueprints still lead to unsatisfactory final experimental results. Additionally, when initializing the blueprint, the size constraint limits the drawing to no more than 20 objects. How was this number determined, and is this setting generalizable?
2. In the ScienceQA-IMG dataset, the open-source LLaVA 13B model achieves an accuracy of 71.6% without fine-tuning, which improves to 88.0% after fine-tuning. The zero-shot method proposed in this paper based on LLaVA achieves an accuracy of 72%, while the best result (based on the closed-source 175B+ GeminiProVision) is 81.1%. This indicates that fine-tuning the 13B model yields significantly better performance than the 175B+ model with carefully designed prompts. In this context, I cannot be convinced by the contribution of this paper.

**Suitability:**

3

---

### Official Review · Reviewer_1XmZ · 2024-05-24

**Rating:** 5
**Confidence:** 2

**Summary:**

This paper has presented an innovative preliminary study that brings multi-agent debate into the scope of multimodal reasoning. And to address two significant challenges in this field: the problem of oversimplifying opinions and diverting focus. the author introduces Blueprint Debate on Graphs (BDoG), a system that restricts debates within a blueprint graph and stores evidence in graph branches, effectively tackling the issues of word-level opinion oversimplification and distraction due to irrelevant concepts. Comprehensive tests are conducted in Science QA and MMBench which confirm BDoG's effectiveness.

**Strengths:**

1. It is a good idea to introduce the graph structure into the multi-agent debate, which not only limits the scope of the discussion to the topic but also makes full use of the structural information of the graph.
2. Extensive experiments are conducted across different MLLMs and datasets.
3. The paper is well organized.

**Limitations:**

1. According to the description in the method section of the article, I think there should be no G->G- connection in Figure 2(e).\
2. Because proponent agents will also enhance the blueprint based on their own insights, there is a question as to whether Proponent agents will introduce irrelevant concepts and cause Focus Diversion problems.
3. BDoG, BDoG^Debate, and BDoG^Graph have different performance gains on Base across different datasets. Can the author make an analysis?

**Suitability:**

3

---

### Official Review · Reviewer_jYmw · 2024-05-27

**Rating:** 4
**Confidence:** 2

**Summary:**

This paper presents a study introducing a novel approach to multimodal reasoning through multi-agent debate. The study addresses two key challenges in existing multimodal reasoning systems: the trivialization of opinions due to excessive summarization and the focus diversion caused by distractor concepts introduced from images. To overcome these challenges, the authors propose a new method called Blueprint Debate on Graphs (BDoG). The experiment results show the superior of BDoG compared with state-of-the-art results in ScienceQA and MMBench.

**Strengths:**

1. Novelty: The introduction of BDoG as a new paradigm and the use of graph-based evidence storage are innovative contributions to the field of multimodal reasoning.
2. Theoretical and Technical Soundness: The formalization of functions and structured debate process ensures the technical correctness of the proposed method.
3. Comprehensive Evaluation: The use of benchmark datasets, comparative analysis, ablation study, and efficiency analysis provides a thorough evaluation of BDoG's effectiveness and efficiency.
4. Clarity and Accessibility: The well-organized structure, detailed explanations, and effective use of visual aids enhance the paper's clarity and accessibility.

**Limitations:**

1. More detailed analysis and experiments are need to introduce why BDoG helps to address trivialization and focus diversion, the paper provides limited empirical evidence showing the complete elimination of these issues.
2. Lack of Implementation Details: Detailed information on algorithms, hyperparameters, and infrastructure is crucial for reproducibility and practical implementation. The github link for the open source is empty.
3.Limited comparative experiments: The experiments are conducted on only two datasets, ScienceQA-IMG and MMBench. While these are standard benchmarks, the inclusion of additional datasets across different domains could strengthen the generalizability claims. For example, testing on datasets related to medical imaging or autonomous driving could provide insights into the method’s versatility.

**Suitability:**

2

---

### Meta-Review · Area_Chair_fdgB · 2024-07-03

**Recommendation:** Accept (Oral)
**Confidence:** 4

**Metareview:**

The paper introduces BDoG, a novel paradigm leveraging graph-based evidence storage to enhance multimodal reasoning. Reviewers highlighted the innovative approach, theoretical soundness, and comprehensive evaluation as significant strengths, noting the structured debate process and thorough experimentation across standard benchmarks. The paper's clarity and effective use of visual aids were also commended. However, reviewers pointed out limitations, such as the need for more empirical evidence on trivialization and focus diversion, detailed implementation information, and broader comparative experiments across diverse datasets. Additionally, concerns were raised about potential focus diversion in proponent agents and the dependency on blueprint quality. Addressing these issues could further validate the contributions and generalizability of BDoG.